# Transcriptomic and Chromatin Landscape Analysis Reveals That Involvement of Pituitary Level Transcription Factors Modulate Incubation Behaviors of Magang Geese

**DOI:** 10.3390/genes14040815

**Published:** 2023-03-28

**Authors:** Jianye Chang, Di Fan, Jiaxin Liu, Yanglong Xu, Xuefei Huang, Yunbo Tian, Jin Xu, Yunmao Huang, Jue Ruan, Xu Shen

**Affiliations:** 1Hubei Key Laboratory of Agricultural Bioinformatics, College of Informatics, Huazhong Agricultural University, Wuhan 430070, China; 2Shenzhen Branch, Guangdong Laboratory of Lingnan Modern Agriculture, Genome Analysis Laboratory of the Ministry of Agriculture and Rural Affairs, Agricultural Genomics Institute at Shenzhen, Chinese Academy of Agricultural Sciences, Shenzhen 518120, China; 3College of Animal Science & Technology, Zhongkai University of Agriculture and Engineering, Guangzhou 510225, China; 4Guangdong Province Key Laboratory of Waterfowl Healthy Breeding, Guangzhou 510225, China; 5College of Life Science, Sun Yat-Sen University, Guangzhou 510642, China

**Keywords:** chromatin accessibility, transcription factor, incubation behavior, geese, ATAC-seq

## Abstract

The incubation behavior of geese seriously affects their egg production performance. Studies on incubation behavior have identified functional genes, but the regulatory architecture relationship between functional genes and chromatin accessibility remains poorly understood. Here, we present an integrated analysis of open chromatin profiles and transcriptome to identify the cis-regulatory element and their potential transcription factors involved in regulating incubation behavior in goose pituitary. Assay for transposase-accessible chromatin sequencing (ATAC-seq) revealed that open chromatin regions increased in the pituitary during the transition from incubation behavior to laying. We identified 920 significant differential accessible regions (DARs) in the pituitary. Compared to the laying stage, most DARs had higher chromatin accessibility in the brooding stage. Motif analysis of open DARs showed that the most significant transcription factor (TF) occupied sites predominantly enriched in motifs binding to the RFX family (RFX5, RFX2, and RFX1). While the majority of TF motifs enriched under sites of the nuclear receptor (NR) family (ARE, GRE, and PGR) in closed DARs at the incubation behavior stage. Footprint analysis indicated that the transcription factor RFX family exhibited higher binding on chromatin at the brooding stage. To further elucidate the effect of changes in chromatin accessibility on gene expression levels, a comparison of the transcriptome revealed 279 differentially expressed genes (DEGs). The transcriptome changes were associated with processes of steroid biosynthesis. By integrating ATAC-seq and RNA-seq, few DARs directly affect incubation behavior by regulating the transcription levels of genes. Five DAR-related DEGs were found to be closely related to maintaining the incubation behavior in geese. Footprinting analysis revealed a set of transcription factors (RFX1, RFX2, RFX3, RFX5, BHLHA15, SIX1, and DUX) which displayed the highest activity at the brooding stage. SREBF2 was predicted to be the unique differentially expressed transcription factor whose mRNA level was down-regulated and enriched in hyper-accessible regions of PRL in the broody stage. In the present study, we comprehensively profiled the transcriptome and chromatin accessibility in the pituitary related to incubation behavior. Our findings provided insight into the identification and analysis of regulatory elements in goose incubation behavior. The epigenetic alterations profiled here can help decipher the epigenetic mechanisms that contribute to the regulation of incubation behavior in birds.

## 1. Introduction

Incubation behavior is an important and specifically evolved reproductive behavior in birds, which is critical for the survival of offspring and thus adult reproductive success. Incubation behaviors lead to ovarian atrophy and the cessation of egg production, which seriously affects the egg production performance of domestic birds. Incubation behavior has been effectively genetically eliminated underlying the high intensity of artificial selection pressures [1]. However, geese still stubbornly maintain incubation behaviors.

Goose reproductive behavior is driven by light stimulation and is tightly regulated by the hypothalamus-pituitary-gonadal axis (HPG) [2,3,4]. The precise coordination of hormonal regulation at all levels of the HPG is essential for the formation of a complete reproductive cycle in chickens. At the neuroendocrine level, proper expression of key reproductive hormones from different pituitary cell types is required for incubation behavior onset and maintenance. The fluctuation of GnRH-FSH/LH and VIP-PRL in the neuroendocrine system determines the formation of the laying and incubation cycle in chickens [5]. High levels of PRL and low concentrations of ovarian steroids are a hallmark of incubation behavior [6,7].

Incubation behavior can be induced via the combined action of estradiol, progesterone, and PRL in ovariectomized female avian [8]. It has been suggested that PRL, estradiol, and progesterone are essential for modulating the reproductive axis for the onset of incubation behavior. However, the molecular role of hormonal signaling is still unclear. The regulatory architecture of incubation behavior and connections between incubation behavior-related gene regulatory networks and behavior maintenance are poorly understood.

Epigenome changes were thought to affect the domesticated phenotype by influencing gene expression in birds [9,10]. The primary regulators of gene expression are transcription factors. Progressively, more and more evidence shows that transcription factors are involved in endocrinology regulation [11]. Transcription factors are the key drivers of cell function and phenotype via their binding to a specific regulatory sequence (cis-regulatory element, CRE). Various elements regulate the actions of genes and alter phenotypic variation [12]. Epigenome alterations lead to heritable phenotypic and transcriptomic changes by regulating transcription factor binding [13,14]. Transcriptome comparisons between laying and brooding chickens have provided potential mechanisms that contribute to incubation behavior. Many transcription factors are activated or inactivated to regulated various genes involved in maintaining low concentrations of gonadotropin and low steroid biosynthesis in the brooding stage, at the pituitary level [15,16]. These reports highlight the key role of transcription factors in the regulation of incubation behavior at transcriptomic and epigenomic levels.

The pituitary gland plays critical roles in regulating many key physiological functions related to growth and reproduction [17]. Previously, our results showed the highly divergent transcriptome in the pituitary between laying and broody stages. Similar results were obtained by others [16,18,19]. The pituitary is a key component of the endocrine system in the modulation of avian incubation behavior. Incubation behavior is related to the cell types transiting on the pituitary [20,21,22]. The differentiation of both lactotrophs and somatotrophs in the pituitary is strictly modulated by transcription factors (TF) [23]. Transcription factors influence the epigenetic priming of cells toward different cell fates [24]. Transcription factors are an important type of gene expression regulator that plays a central role in regulating various physiological processes such as pituitary cell fate and the secretion of cell-type-specific pituitary hormones [23,25].

Gene expression is controlled by accessible chromatin. Genes with chromatin accessibility are more likely to be regulated by TF and are differentially expressed at the mRNA level. We suspect that the potential mechanism of hormonal signaling in the formation of incubation behavior is related to chromatin accessibility. To characterize the chromatin state landscape of the pituitary of brooding geese, integrative transcriptome and chromatin landscape analysis was performed to identify cis-regulatory elements and their potential transcription factors involved in regulating goose incubation behavior. These results may provide new evidence concerning the correlations among chromatin accessibility, gene expression, and hormone signaling for goose incubation behavior.

## 2. Materials and Methods

### 2.1. Ethics Statement

This study was carried out in strict accordance with the Experimental Animal Committee of Zhongkai University of Agriculture and Engineering (NO. 20201226012). All efforts have been made to minimize animal suffering.

### 2.2. Animals and Sample Collection

The Chinese natively bred Magang goose was obtained from the Wen Jianmin goose farm (Panyu, Guangzhou, Guangdong, China). We used twelve female geese in the present study; six were brooding (broodiness group) and the other six were continuously laying (laying group). Pituitary samples were collected, and transcriptome sequencing (RNA-seq) was performed (*n* = 3 for each stage). Assays were taken for transposase-accessible chromatin with high-throughput sequencing (ATAC-seq) (*n* = 2 for each stage). Serum samples were collected via venipuncture (*n* = 6 for each stage) and then stored at −80 °C before the next step. A section of the ovary was obtained and stained for observation (*n* = 6 for each stage).

### 2.3. ATAC-seq Library for Frozen Pituitary Tissues

Nuclei were isolated from frozen pituitary according to the Ryan Corces protocol [26]. The frozen pituitary was placed into a pre-chilled 2 mL Dounce tissue grinder set (Cat. No. D8938-1SET) (Sigma-Aldrich, Darmstadt, Germany) containing 2 mL cold 1× HB and then thawed for 5 min. The tissue was filtered during transfer using a 70 μm cell strainer (Cat. No. 431751) (Corning, New York, NY, USA), and the homogenate was transferred to a pre-chilled Eppendorf 2 mL Lo-Bind tube (Cat. No. Z666556-250EA) (Sigma-Aldrich, Darmstadt, Germany). Pellet nuclei were generated by spinning the homogenate for 5 min at 4 °C at 350 RCF in a fixed-angle centrifuge. Iodixanol (Cat. No. D1556-250 mL) (Sigma-Aldrich, Darmstadt, Germany) was used for collecting the nuclei. Isolated nuclei were counted with trypan blue staining in a cell counter, and 30 K high-quality nuclei were used to prepare an ATAC-seq library.

This ATAC-seq library was prepared following the instructions of the TruePrep DNA Library Prep Kit V2 for Illumina (Vazyme, Nanjing, China). Nuclei pellets were incubated in a 50 µL transposition mix (10 µL 5 × TTBL buffer, 5 µL TTE Mix, 35 µL ddH2O) for 30 min at 37 °C. Transposed DNA was then purified with VAHTS DNA cleaning beads (Vazyme, Nanjing, China) and then amplified for 12~15 cycles with PCR. Libraries were purified and assessed for fragment length distribution with a Bioanalyzer Qseq 100 Bio-Fragment Analyzer (Bioptic Inc., New Taipei City, Taiwan) and submitted for paired-end 150 bp sequencing on the NextSeq platform.

### 2.4. ATAC-seq Data Analysis

The Fastp [27] (version 0.23.1) software was used to remove adapter sequences, poly-X in 3′ ends, and reads that had a phred-scaled quality score of less than 15 for more than 15% of the bases. The trimmed fastq files were mapped to the *Anser cygnoides* genome obtained from the NCBI database (AnsCyg_PRJNA183603_v1.0) using Bowtie2 [28] (version 2.3.4 with named parameters ‘--very-sensitive -X 2000′. The sequence alignment map (SAM) files were compressed to the binary alignment map (BAM) version on which Samtools [29] (version 1.7) was used to filter reads that were unmapped, mate unmapped, not primary aligned, or failed platform quality checks. Read pairs mapped to mitochondria DNA were discarded using Samtools (version 1.7). Redundancy read pairs from PCR amplification were also removed afterward using Picard [30] MarkDuplicates (version 2.18.29). The Deeptools [31] alignmentSieve (version 3.1.3) software was used to move the forward chain of the BAM file forward by 4 bp and the reverse chain backward by 5 bp with parameter ‘-ATACshift’. Then, open accessible-regions for each shifted bam file were defined by the peaks called by MACS2 [32] (version 2.1.1) with the parameters “-g ‘genome_size’ -f BAMPE –q 0.05 --keep-dup all”. Motif analysis on peak regions was performed by Hypergeometric Optimization of Motif Enrichment [33] (HOMER version 4.9.1), using the function findMotifsGenome.pl.

### 2.5. The Identification of DARs and Peak Annotation

Analysis of ATAC-seq for differential accessibility was carried out in the DiffBind [34] package (version 3.6.5) using the default threshold parameters of |Fold| ≥ 0.6 and *p*-adj ≤ 0.01 by DESeq2. Annotations of the peaks were achieved using the HORMER annotate function.

### 2.6. Computational Footprint Analysi

The non-redundant vertebrate TF motifs were downloaded from the JASPAR database (release 2020) [35]. Computational footprint analysis was conducted across each merged bam file using TOBIAS [36]. Briefly, the Tn5 transposase sequence preference of cutting sites was estimated and corrected using the parameter of ‘ATACorrect’. The deletion of ATAC-seq signals given rise from protein binding and the neighboring signals around binding sites were calculated using the parameter of ‘FootprintScores’. The differential binding TFs were detected using the parameter of ‘BINDetect’. All transcription factors with −log10 (*p*-value) above the 95% quantile or differential binding scores smaller/larger than the 5 and 95% quantiles were considered differential binding TFs.

### 2.7. RNA-seq and Differential Gene Expression Analysis

Total RNA extracted as described above was used for transcriptome sequencing on Illumina HiSeq 4000 to generate 150 bp paired-end reads. Clean reads filtered by Fastp software were mapped to the goose genome (AnsCyg_PRJNA183603_v1.0) using STAR 2.0.4 software with default parameters, DESeq2 [37] was performed to identifyDEGs. The threshold level was set with a standard: FDR ≤ 0.05, |log2 fold change| ≥ 1.0. Each pairwise combination of the two reproductive stages was investigated. Real-time PCR was used to validate the differential expression genes.

### 2.8. GO and KEGG Pathway Enrichment Analyses

The gene ontology (GO) enrichment analysis of genes was performed using Metascape (https://metascape.org/, accessed on 1 May 2022) [38]. KEGG enrichment analysis was performed using KOBAS [39] with a corrected *p*-value cutoff of 0.05 to judge statistically significant enrichment.

### 2.9. Ovarian Histology and Serum Hormone

The geese ovaries were collected immediately after slaughter. The dissected ovarian tissues were immediately fixed in paraformaldehyde. The slides were stained by Hematoxylin and Eosin (H&E) to investigate the structure, morphology, and histological changes. Serum samples were isolated from blood samples. The concentrations of PRL and LH in serum were measured by Enzyme-Linked Immunosorbent Assay (ELISA), according to the instructions of the Prolactin ELISA kit (Cat. No. EK18156) (SAB, Baltimore, MD, USA) and LH ELISA kit (Cat. No. EK11984) (SAB, Baltimore, MD, USA). Statistical analysis was performed by the Analysis of Variance (ANOVA). A probability level of *p* < 0.05 (*) or *p* < 0.01 (**) was used to indicate significance.

## 3. Results

### 3.1. Ovarian Histology Structure and Hormone Concentration

The ovarian cortex of laying geese contains a hierarchy of all stages of developing follicles, five to eight of the largest hierarchy of preovulatory follicles are present, pre-hierarchical follicles including small yellow follicles (SYF), large white follicles (LWFs), and small white follicles (SWFs), as well as the post-ovulatory follicles (POF) are visible, whereas the ovaries manifested as regression and atresia in the brooding stage geese. In addition, ovarian histology of the laying geese and broody geese were investigated. Consistent with previous results, many primary follicles and secondary follicles can be observed in the ovaries of the laying stage; however, the brooding geese had numerous primary follicles (Appendix A). We measured serum PRL and LH concentrations of individuals in both stages. Compared to the laying stage, the concentration of PRL in the serum was significantly higher in the brooding stage (*p* < 0.01) (Appendix A), while a significantly lower concentration of LH was observed in the serum (*p* < 0.05) (Appendix A).

### 3.2. Characteristics of Chromatin Accessibility in Goose Pituitary Underlying Two Reproductive Stages

We isolated the nuclei from the geese pituitary and investigated the chromatin accessibility involved in incubation behavior. We profiled the chromatin accessibility regions in both the laying stages and the brooding stages (Figure 1). We generated more than 48 million paired-end reads for each sample. The Q20 ratio, Q30 ratio, and GC content in the clean data were calculated, which ranged from 95.2% to 98.1%, 92.2% to 94.6%, and 46.2% to 47.8%, respectively, 94.6 to 96.2% of reads from all libraries were mapped to the goose genomes from NBCI (Appendix A).

Pearson correlation analysis and PCA analysis of ATAC-seq showed the profiles of chromatin accessibility in the pituitary were strikingly divergent between laying stage and brooding stage geese (Figure 2a and Appendix A). A distribution plot of insert fragment length clearly shows that nucleosome packing and chromatin sequence insertion exhibit a distinct periodicity of approximately 200 bp in the chromatin landscape of the pituitary gland, which is the normal insertion fragment pattern for the ATAC-seq library (Figure 2b,c and Appendix A). The majority of reads were located in regions less than 100 bp, indicating that the nucleosome-free open regions were cut and sequenced. Mononucleosome, dinucleosome, and trinucleosome patterns were observed. We also assessed the quality of all libraries based on the peak signal distributions and the transcription start site enrichment (TSS) score (Appendix A).

Open chromatin regions can generally combine specific transcription factors to affect the transcription process. We investigated the differences between transcription factors in the pituitary during the laying and brooding stages at a genome-wide level. For comparison, we first calculated the proportion of genomic positions of peaks in each library and identified 9174 and 7155 differentially accessible regions of chromatin by merging results from the laying stage and the brooding stage, respectively. (Figure 2d and Appendix A). Our data revealed that the profile of chromatin accessibility was strikingly different between the laying stage and the brooding stage. Then, we conducted motif enrichment analysis on the consensus peaks under the two states.

The motif analysis suggested that the ranking of motif usage is a little different in the pituitary between the laying stage and the brooding stage. As expected, a significant enrichment of the binding motif of CTCF, SIX2, SIX1, BORIS, RORIN, and helix-turn-helix (HTH) superclass (RFX, RFX2, RFX1, and X-box) was identified in the laying stage; likely reflecting that these transcription factors are the master regulator responsible for maintaining the physiological state of laying in pituitary (Figure 2e). However, transcriptional regulatory regions with motif binding of CTCF, BORIS, Homeobox families (SIX2, SIX1), NR family (GRE, PGR, ARE), and HTH families (RFX, RFX2) showed open chromatin accessibility in the brooding stage (Figure 2); the detailed information on all transcription factors identified is presented (Appendix A). The observations for enrichment motif showed the epigenetic landscape of the pituitary is stable in both reproductive stages, suggesting that open chromatin regions are required for essential pathways to maintain a multitude of the endocrine function of the pituitary in geese. Thus, further analysis of the data is required to filter and identify key chromatin regions that regulate incubation behavior. In addition, we showed the trace near the TRHR gene, indicating the authenticity and repeatability of our data (Figure 2).

### 3.3. Differential Chromatin Accessibility Regions between Two Reproductive Stages

To identify the potential cis-regulatory elements in the pituitary at the chromatin level related to incubation behavior, we compared chromatin accessibility in the pituitary between laying and brooding. A total of 920 significant differential accessible regions (DARs) were identified between the laying and brooding pituitary using DiffBind (abs (Fold) ≥ 0.6, FDR ≤ 0.01). These DARs were annotated into 847 genes based on their distance to the nearest TSSs. The DARs were clearly classified into two distinct clusters based on the accessibility in each fraction based on the K-means clustering method (Figure 3a). The two clusters revealed strikingly divergent chromatin accessibility between the two stages. Cluster 1 (114 peaks) showed a higher average chromatin accessibility in the laying samples, while cluster 2 (*n* = 806) showed higher accessibility in the broody samples (Figure 3a).

Based on the genome-wide functional regions, the majority of up-regulated peaks were mapped to intronic regions, followed by intergenic regions, and promoter regions; while the down-regulated peaks mostly come from intergenic regions, intronic, and exon regions of genomes. There was no significant proportional change between up-regulated and down-regulated DARs in the intergenic and promoter-TSS regions. However, the proportion of down-regulated DARs, in both exon and TTS regions, was significantly increased, whereas the proportion of up-regulated DARs was relatively decreased. On the contrary, a higher proportion of peak up-regulation occurred in intron regions, which may suggest that enhancers are an important regulatory layer in the broody stage (Figure 3b).

We then compared the frequencies of the transcription factor binding motifs at differentially accessible regions between the two stages by using homer software and calculated the *p*-values and enrichment score of the motifs. Here we refer to the peaks in cluster1 as closed DARs, and the peaks in cluster2 as open DARs. Motif analysis of open DARs at the incubation behavior stage showed that the most significant transcription factor occupies sites enriched in motifs for the HTH superclass (RFX5, RFX2, RFX1, RFX and X-box), homeobox (six2, six1, Lhx3, LHX9, and EN1), NF1, and NeuroD1 in open DARs, while an enrichment of motifs for NR (ARE, GRE, and PGR), homeobox (Bapx1), EFL-1,AT3G10030,ERF115, LIN-15B, and MYB (MYB51, MYB40, and MYB39) were identified in the closed DARs at the incubation behavior stage (Figure 3c). The motif enrichment analysis in DARs reveals a set of potential TFs highly related to incubation behavior in the pituitary.

To explore the potential function of DARs, GO enrichment analysis was performed for DARs related to genes. GO annotation of biological processes related to open DARs showed highly enriched neuron projection development and neuronal differentiation in the brooding stage (Figure 3e); however, closed DARs related to genes were strongly enriched in negative regulation of transport and positive regulation of protein phosphorylation in biological processes (Figure 3).

### 3.4. Transcriptome Change in Pituitary Underlying Two Reproductive Stages

To assess the potential impact of chromatin changes on gene expression, transcriptome sequencing was conducted to investigate the differentially expressed genes (DEGs) in the pituitary underlying the two reproductive stages (Figure 1 and Appendix A). Principal component analysis (PCA) of the normalized count data separated the sample of the two stages (Figure 4a). We identified 225 DEGs (Fold Change ≥ 1.5, FDR ≤ 0.05) between the laying stage and the brooding stage, and 102 genes were upregulated in the brooding stage, while 123 genes were down-regulated (Appendix A). The functional genes involved in incubation behavior, such as TSHR and SREBF2 were markedly down-regulated in the brooding pituitary [15]. A bar plot showed the top 10 DEGs ranked by the FPKM value of the fold change (Figure 4b). Remarkably, the prolactin gene PRL was the most highly expressed gene in the pituitary gland and showed a clear and consistent difference between the two states (log2FoldChange = −1.53, FDR = 0.009). qPCR was performed to validate the gene expression difference (Appendix A).

GO enrichment analysis revealed that up-regulated DEGs were associated with lipid transfer activity and transporter activity in molecular function (Figure 4c), and down-regulated DEGs were associated with monooxygenase activity, glycosyltransferase activity, and oxidoreductase activity (Figure 4d). Interestingly, the GO analysis result indicated that the up-regulation genes and down-regulation genes are linked to lipid metabolism in the biological process and play different roles in different processes of lipid metabolism. KEGG analysis revealed that the highlighted pathways are significantly overrepresented in steroid biosynthesis, metabolic pathways, and glycerophospholipid metabolism (Figure 4e). Our results indicated that the alteration of lipid metabolism in the pituitary participated in the regulation of incubation behavior.

### 3.5. Expression Change in DAR-Related Genes to Goose Incubation Behavior

In our study, the pituitary gland showed comparatively less variation at the level of gene expression and more change involving chromatin accessibility in goose incubation behavior. To determine the effects of chromatin accessibility on gene expression, no significant correlation was observed. We performed overlapping analysis of DARs and DEGs. Change in chromatin accessibility regulated a small set of DEGs. Five genes were both differently expressed and showed differential chromatin accessibility between the laying stage and the brooding stage (Figure 5, Appendix A). LOC106036132, OTOGL and PCDH18 showed increased expression and altered chro-matin accessibility during incubation. The expression levels of PRL and GPX3 genes were higher during the laying period, but chromatin accessibility was decreased. The ex-pression and chromatin landscape of the above five genes are shown in Appendix A.We next investigated the transcription factors within open chromatin regions of DAR-related DEGs in the goose pituitary and found SREBF2 was the unique differentially expressed transcription factor bound to DARs in the brooding stage (Appendix A). SREBF2 were significantly down-regulated and enriched in hyper-accessible regions of PRL in the broody stage (Appendix A).

### 3.6. Key Transcription Factors Were Explored by Footprinting Analysis

We performed footprinting analysis to estimate the transcription factor (TF) binding on different open chromatin regions. We collected the vertebrate TF binding profiles from the JAPSAR database [35]. After correction for deviation, factors with −log10 (*p*-value) above the 95% quantile or differential binding scores smaller/larger than the 5 and 95% quantiles were defined as differentially bound TFs between the laying stage and the brooding stage (Figure 6). The differentially bound TFs are labeled in the Figure 6. We observed that TFs exhibit similar binding dynamics, as TFs from the same family often have highly similar binding motifs. These transcription factors may play an important regulatory role in the maintenance of incubation behavior. Interestingly, we compared the TF binding between the laying stage and the broody stage in the pituitary. Footprints of a set of transcription factors (RFX1, RFX2, RFX3, RFX5, BHLHA15, SIX1, and DUX) displayed the highest activity at the brooding stage, suggesting that factors are likely involved in incubation behavior (Figure 6).

## 4. Discussion

Magang geese are typical short-day breeders, which have strong incubation behavior [5]. Ninety percent of female geese present incubation behavior after laying one clutch of eggs with a high incubation constancy and a long incubation duration time. The seasonal breeding characteristics and the poor laying performance caused by incubation behavior have severely restricted the development of the waterflow industry. Over the past 20 years, avian incubation behavior has been studied extensively at the molecular level, with many previous reports focusing on identifying functional genes and genetic mutations associated with this behavior [4,40,41,42,43]. In spite of these major advances, the underlying mechanisms remain unknown as incubation behavior is a low heritability trait. Incubation behavior had been demonstrated to be induced by the combined action of three hormones [8], suggesting that changes involved in incubation behavior many occur at different levels, such as transcription and post-transcriptional regulation. The transcriptome is a dynamic component that plays a critical role in determining phenotypes. Transcriptome profiling of the HPG axis has revealed distinct expression patterns of gene and non-coding RNA associated with incubation behavior [15,16,44]. However, the genetic architecture and epigenetic regulatory mechanisms of incubation behavior is still unclear.

In the present study, we comprehensively profiled the chromatin accessibility and transcriptome in the pituitary tissue of both laying and brooding geese. Accumulating evidence has emerged showing that changes in chromatin accessibility are associated with their regulatory role in the endocrinology function of the pituitary gland [17,23,45]. Although alterations in chromatin accessibility might be related to different expression patterns between the laying stage and the brooding stage, how these alterations relate to phenotype remains unclear. Transcription factors influencing gene expression can be modulated by chromatin accessibility at transcription factor binding sites. We previously reported that the transcription factors SREBF2, PGR, and SF1 act as central signal modulators during the transition from laying to brooding at the molecular level [15]. We investigated the characteristics of open chromatin regions in the pituitary gland underlying both the laying stage and the brooding stage in terms of chromatin level; the open chromatin region showed alterations in chromatin accessibility preferentially occurring during the brooding stage. As expected, CTCF was enriched at the two reproductive stages. Our results showed that the nuclear receptor (NR) superfamily (GRE, PGR, and ARE) was significantly enriched at the brooding stage. Transcription factors of the NR superfamily were involved in a numbers of physiological processes such as reproduction, circadian rhythms, and metabolism [46]. Most members of the NR superfamily are activated by hormones, such as thyroid hormones, steroids, vitamin D, or retinoic acid [47]. Three transcription factors, glucocorticoid response elements (GRE), progesterone receptors (PGR), and androgen response element (ARE), were strongly enriched in the brooding stage. Glucocorticoid release is a classic endocrine response to stress, which can directly modulate reproductive function in the HPG axis [48]. In quail, GRE corticosterone treatments increased the expression level of GnIH through GRE, which is located in the GnIH promotor [49]. ARE is the target binding element present in promoters or enhancers of genes targeted by the androgen receptor (AR) [50]. AREs can function as glucocorticoid or progesterone response elements and vice versa as steroid hormone receptors are all similar in their receptor structures [51]. PGR belongs to steroid receptors and plays a key role in avian reproduction. Progesterone regulated chicken ovulation and incubation behavior via the pituitary [8]. PGR is a nuclear receptor transcription factor which is essential for female reproduction and plays a pleiotropic role in different tissues of the female reproductive systems [52]. In the female reproductive systems, PGR transcriptional regulation is highly diverse and tissue-specific and is related to different physiological roles in different target tissues [53]. The expression level of PGR significantly declined at the brooding stage. We suspected that PGR acted as a central signal modulator during the transition from laying to brooding via the pituitary [15].

Phenotypic changes mostly result in the activity of cis-regulatory DNA elements, driven by changes in transcription factor (TF) binding [54]. At the chromatin level, we identified 920 DARs related to the reproductive stage. The differentially open accessible regions in the brooding stage are enriched for non-promoter regions, with binding motifs of 153 transcription factors enriched. The RFX HTH family (RFX2, RFX, Rfx1, and Rfx5), the sine oculis (six) homeobox family (six2 and six1), the NeuroD bHLH family (NeuroD1 and NeuroD2), the Lhx family (LHX3 and LHX9), NF1, and EN1 were the significantly enriched in open DARs at the incubation behavior stage. The RFX HTH family are highly conserved in animals and act as master regulators of central nervous system development and ciliogenesis [55,56,57]. RFX transcription factors regulate their target genes through X-box in the promoter and are highly expressive in four organ systems: the immune system, gastrointestinal tract, reproductive system, and nervous system. RFX2 was reported as a key regulator involved in mouse spermiogenesis [58,59,60]. Through footprint analysis, similar to six1 and six2, RFX1 and RFX2 had a higher footprint score in the brooding stage. RFX5 was an essential and highly specific regulator of major histocompatibility class II (MHCII) gene expression in the immune system. Members of the six homeobox transcription factor family play an important role in organogenesis and differentiation in a wide range of animal species [61]. NeuroD transcription factors govern photoreceptor genesis and regeneration, while NeuroD knockdown prevents cell cycle exit and photoreceptor regeneration in the retina of mice [62]. NeuroD1 regulates the thyroid hormone receptor β2 and cone opsins in the developing mouse retina [63]. NeuroD1 is strongly expressed in the gonadotrope progenitor, and over-expression of NeuroD1 increases the mRNA expression level of GnRH receptor gene in mouse gonadotrope cells [64]. LIM homeodomain transcription factors are required for pituitary gland and nervous system development. In the pituitary, LHX3 is reportedly involved in the activation of the FSH β-subunit gene gonadotrope cell [65]. Lhx9 plays an important role in the regulation of cell proliferation and migration, which is essential for mouse gonad formation [66]. The role of NF-1 in the pituitary seems to be as a repressor related to human growth hormone [67].

However, the NR superfamilies (ARE, GRE, and PGR) were significantly enriched in closed DARs at the incubation behavior stage. The transcriptional activity of NRs is modulated by various ligands, including hormones and lipids [68]. By sensing changes in lipid metabolite levels, NRs drive differential gene expression producing different physiologic effects [46]. Androgens start puberty and play a role in reproduction. Pituitary androgen receptor signaling is previously reported to regulate prolactin release in males [69]. ARE motifs were present in the promoters or the enhancers of genes targeted by the androgen receptor. Previous studies showed that androgen receptors positively regulated gonadotropin-releasing hormone receptors in pituitary gonadotropes [70]. The glucocorticoid receptor is a steroid hormone-activated transcription factor that regulates the transcription of thousands of target genes by binding to GRE upstream of target genes, thus producing different physiological functions [71]. Progesterone plays critical roles in reproduction across vertebrates. Nuclear PGR is a ligand-dependent transcription factor responsible for mediating progesterone action related to reproduction [72]. Our results indicated that closed DARs related to the transcriptional activity of NRs, which sense change among androgen, glucocorticoid, and progesterone in the pituitary.

Conversely, the majority of differential gene expression between the laying and brooding stages were largely similar to those reported in previous bird studies in transcriptome level [15,16]. Cholesterol metabolic processes were significantly down-regulated in the laying stage. To the best of our knowledge, high levels of prolactin are one of the recognized characteristics of incubation behavior. However, in our data the expression pattern of PRL significantly declined in the brooding stage; notably, the serum concentration of the PRL hormone was higher in the stage of broodiness than in the laying stage, which is consistent with the observations in other breeds of geese [73]. We suspect that the brooding geese, deprived of their eggs and nests, exhibited a significant reduction in the mRNA level of PRL [45,74,75]. ATAC-seq showed that chromatin changes in intergenic regions of the PRL gene. Compared to the laying stage, steroid biosynthesis, metabolic pathways, and glycerophospholipid metabolism pathways were all significantly changed, which is consistent with previous results for birds [15]. SREBF2 is the central player of lipogenesis, as it is the master regulator of cholesterol synthesis [76]. In our study, SREBF was the differentially expressed transcription factor whose mRNA expression level was significantly down-regulated and enriched in hyper-accessible regions of PRL during the broody stage. Moreover, the lipoprotein metabolism gene is rarely upregulated during the brooding stage. Cholesteryl ester transfer protein (CETP) was the most expressed gene in the brooding stage, indicating that lipoprotein metabolism was the most dynamic change between brooding and laying.

The integration of ATAC-seq and RNA-seq results showed that chromatin changes are uncoupled from differential gene expression. We observed five gene changes in both chromatin accessibility and expression level. GPX3, which is known to be epigenetically regulated, showed significant down-regulation of mRNA expression levels, and the chromatin accessibility was altered. Loss of Gpx3 induces oxidative stress and increases prostatic intraepithelial neoplasia proliferation in prostatic cancer [77], and silencing GPX3 expression has been reported to enhance metastasis of human thyroid cancer. DAR-related DEGs, which have low expression in the brooding stage, demonstrated their function on the reproductive system. PCDH18 is thought to play a role in cell–cell connections in the brain, and haploid defects in this gene may lead to alterations in brain development and associated malformations [78]. OTOGL mutations have been found to be associated with hearing loss [79]. In addition, correlation analysis of our nine-quadrant plot shows no significant correlation between chromatin accessibility and gene expression level.

We downloaded the motif probability matrix corresponding to transcription factors from the Jaspar database for evaluating the activity of transcription factors at different stages. Four RFX transcription factors (RFX1, RFX2, RFX3, and RFX5), BHLHA15, SIX1, and DUX4 were highly active in the brooding phase, suggesting that these factors are likely involved in goose incubation behavior. The RFX transcription factor family was identified as the highly active transcription factor in the brooding stage in our footprint analysis with a significantly up-regulated chromatin accessibility. This finding is very interesting because several previous studies have shown that the RFX transcription factors play a crucial role in the ciliogenesis involved in the spermatogenesis [58,59,60].TFs binding footprints have been observed in some promoter regions. Further footprint analysis will help reveal the dynamics of open chromatin regions and chromatin accessibility of these genes and identify cis-regulatory elements associated with their specific transcription during the transition phase of incubation behavior. These potentially cis-regulatory elements provide candidates for experimental verification of protein–DNA interactions for research into the genetic breeding of poultry.

## 5. Conclusions

In this study, we applied ATAC-seq and RNA-seq to identify the key DARs and DEGs in the pituitary that affect the incubation behavior of geese. We first identified 920 DARs and important transcription factors that might regulate the emergence of incubation behaviors by ATAC-seq from pituitary tissues of laying geese and broody geese. We obtained genes significantly related to steroid biosynthesis in the transcriptome. In addition, the combined analysis of ATAC-seq and RNA-seq identified five genes closely related to the incubation behavior of geese; SREBF2 may bind to the hyper-accessible region of the PRL gene and affect its expression changes. Finally, footprinting analysis revealed a set of transcription factors that displayed the highest activity at the brooding stage. Taken together, we provide a new insight into the regulatory elements of goose incubation behavior from a pituitary perspective, help reveal epigenetic mechanisms regulating bird incubation behavior, and provide a new clue for genome-assisted breeding in the poultry industry.

## Figures and Tables

**Figure 1 genes-14-00815-f001:**
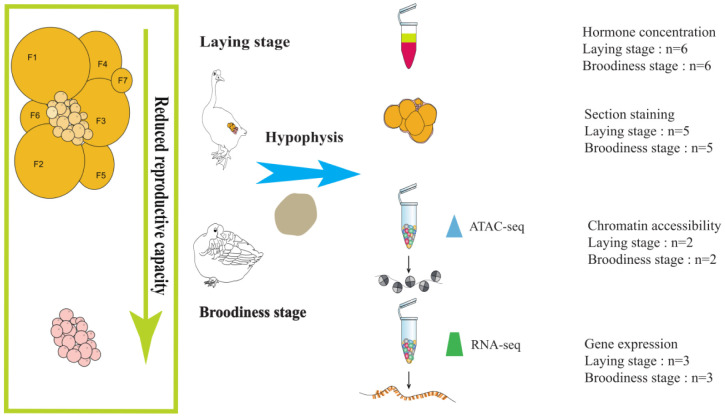
Schematic diagram of experimental design. Ovarian tissue from the laying and broodiness groups was extracted to make sections and stain for hematoxylin–eosin staining (H&E staining) (*n* = 5). Serum samples of pituitary tissue from the laying and broodiness groups were extracted for hormone concentration determination (*n* = 6). Nuclei were extracted from pituitary tissues from the laying and broodiness groups for ATAC-seq (*n* = 2), and RNA-seq (*n* = 3). “n” represents the sample size in each experiment. F1 to F6 means the largest hierarchy of preovulatory follicles present in laying stage of geese.

**Figure 2 genes-14-00815-f002:**
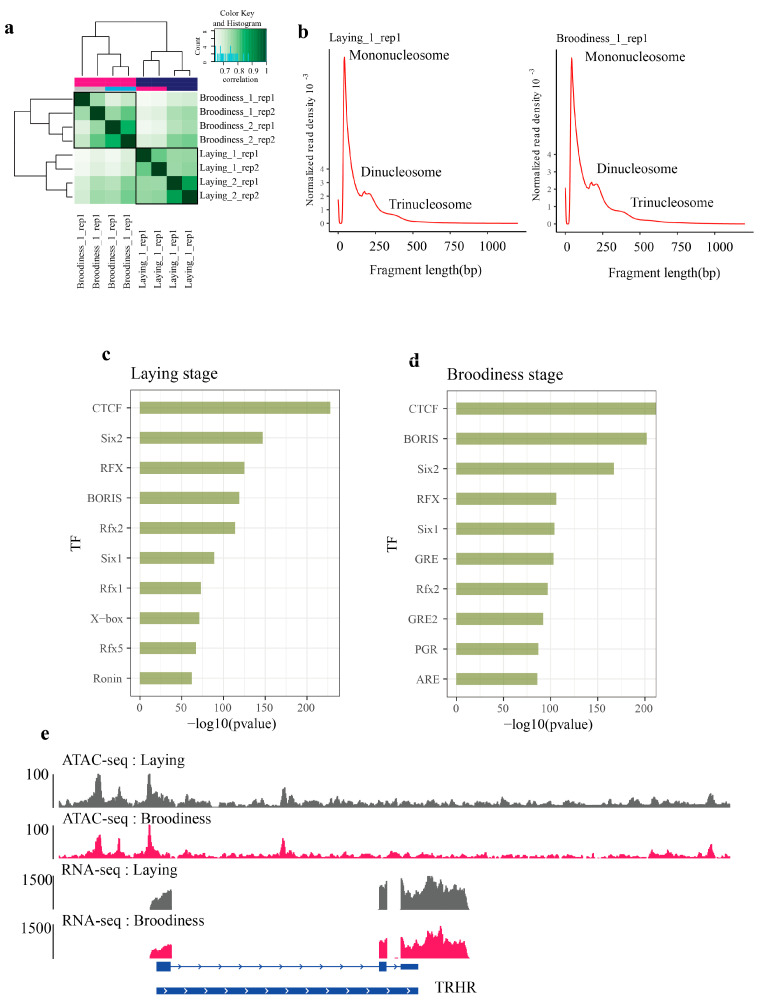
The quality control of ATAC-seq and characteristic of chromatin accessibility in goose pituitary at the genome-wide level. (**a**) The Pearson correlation heat maps show correlations between all ATAC-seq libraries, with biological replicates at laying and broodiness stages clustered together separately and with high consistency between technical replicates. (**b**) The curve graph shows the number of length changes in insertion fragments in ATAC-seq library. The highlighted peaks represent mononucleosome, dinucleosome, and trinucleosome, respectively, showing the standard pattern of length of insertion fragments in ATAC-seq library. (**c**) The histogram shows transcription factors corresponding to motif enrichment in open chromatin regions of the whole genome in laying stage, *X*-axis represents −log (*p*-value) and *Y*-axis represents transcription factors. (**d**) The histogram shows transcription factors corresponding to motif enrichment in open chromatin regions of the whole genome in the broodiness stage. (**e**) Peak diagram of the relationship between open chromatin region near TRHR and TRHR gene expression. The pink is the broodiness stage and grey is the laying stage.

**Figure 3 genes-14-00815-f003:**
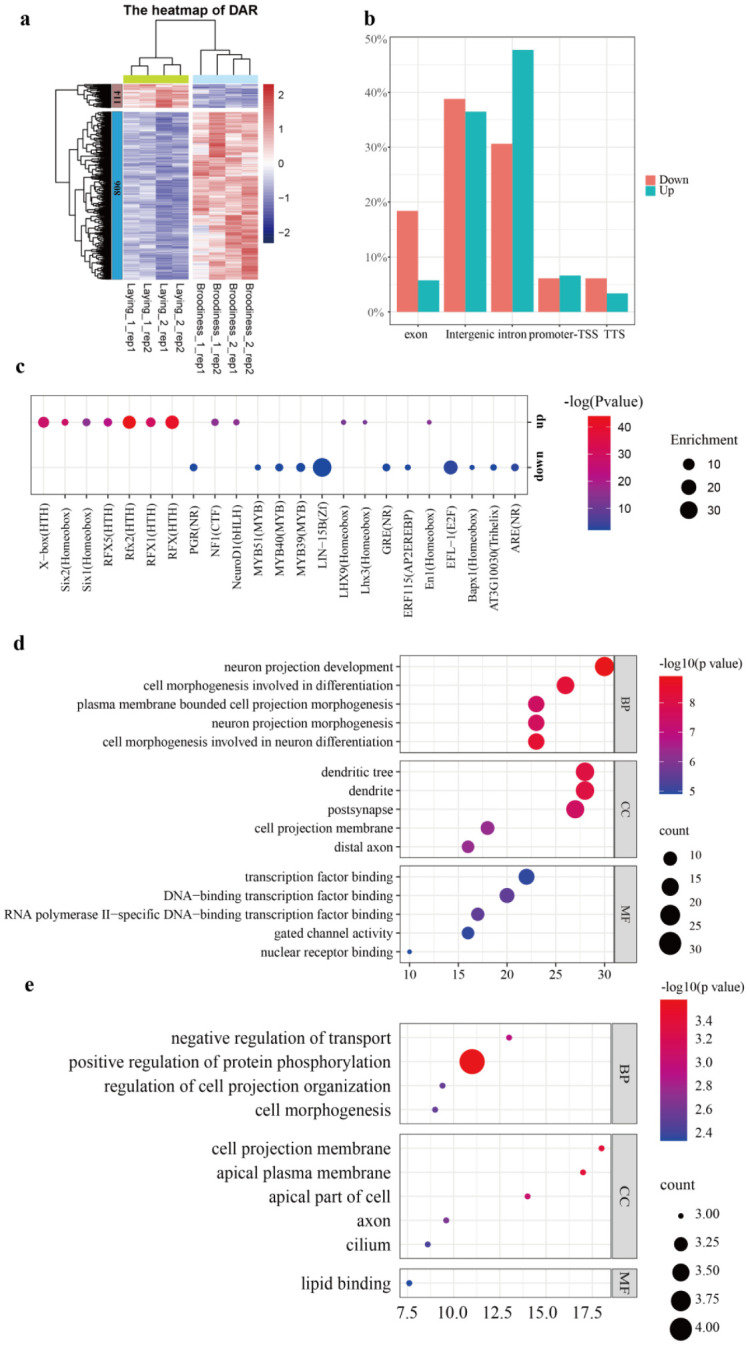
Calculation, distribution annotation, and GO function annotation of differential accessibility regions (DARs) in ATAC-seq data. (**a**) The heatmap of the 920 DARs between pairwise comparisons, 806 of which had increased chromatin accessibility in the brooding samples and 114 of which had increased chromatin accessibility in the laying samples. (**b**) The histogram shows and compares the distribution of up-regulated and down-regulated DARs on the genome. (**c**) The bubble plot shows the first 11 transcription factors corresponding to the motifs enriched in the DARs sequence, where the bubble color value is −log (*p*-value), representing the significance of the motifs, and the bubble size value is the enrichment score calculated by dividing target by background. (**d**) GO function analysis of annotated up-regulated and down-regulated differential accessibility regions. (**e**) GO function analysis for the up-regulated and down-regulated differential accessibility regions.

**Figure 4 genes-14-00815-f004:**
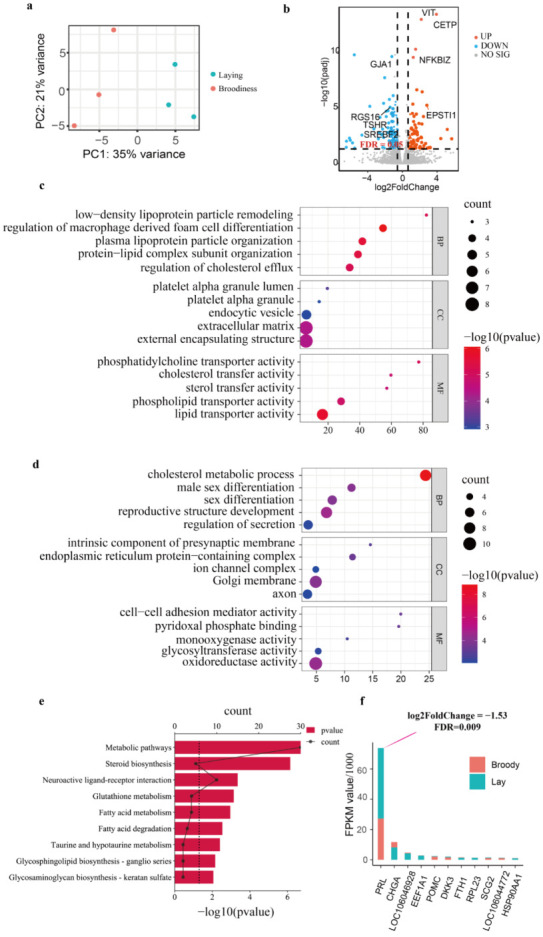
Transcriptome analysis of the differentially expressed genes (DEGs) in the goose pituitary underlying two reproductive stages. (**a**) The principal component analysis of different libraries from two reproductive states showed good transcriptome data quality (blue dot, laying stage; red dot, broodiness stage). (**b**) Volcanic plot of differential gene expression, several interesting genes are labeled (blue dot: down, red dot: up, gray dot: no significant difference). (**c**) GO (BP, CC, and MF) analysis of up-regulated DEGs. (**d**) GO (BP, CC, and MF) analysis of down-regulated DEGs. (**e**) KEGG pathway analysis of all differential genes, where the transverse bar represents −log10 (*p*-value) and discounting represents the number of genes present in pathway terms. (**f**) The stacked plot of top ten genes with the highest FPKM values, and the log2FoldChange and FDR of PRL genes were labeled.

**Figure 5 genes-14-00815-f005:**
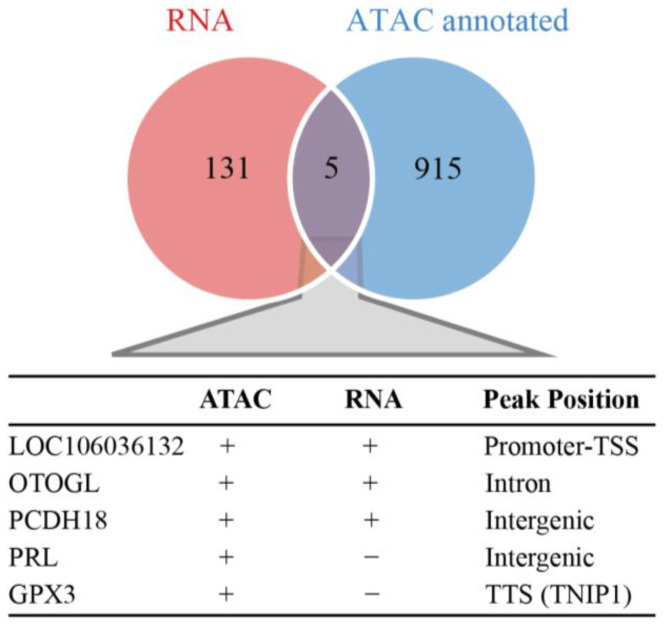
Joint analysis of ATAC-seq and RNA-seq. The above Venn plot shows the overlap analysis of differentially open regions and differentially expressed genes. The table below shows the differential changes in five overlapping genes in the two omics, details about openness and gene expression can be observed in the table.

**Figure 6 genes-14-00815-f006:**
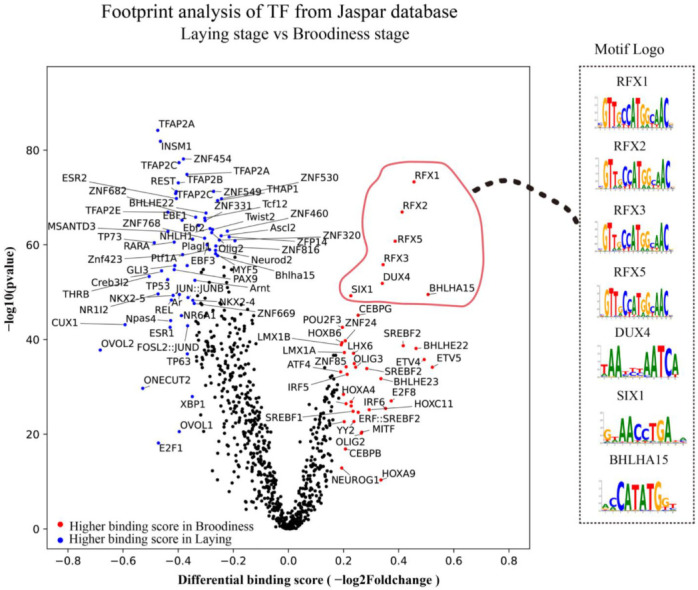
Pair-to-pair comparison of TF activity at different stages of reproductive development. Volcano plot shows differentially binding activity of −log10 (*p*-value) (all provided by TOBIAS software) for all TF motifs investigated; each point represents a motif. All TFs with −log10 (*p*-value) above the 95% quantile or differential binding scores smaller/larger than the 5% and 95% quantiles (top 5% in each direction) were considered differential binding TFs. For the laying stage, specific TFs are marked in blue, while specific factors of the broodiness stage are marked in red. Seven prominent examples from specific TFs in the broodiness stage are selected and illustrated on the right.

## Data Availability

The data sets supporting the results of this article were included within the article and additional files. Raw sequencing reads are available at Genome Sequence Archive (GSA) database of the National Geoscience Data Centre (NGDC) (bio-project accession: PRJCA012997, ATAC-seq accession CRA008881, and RNA-seq accessions CRA008882).

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
