# Peer review of "Transcriptomic and Chromatin Landscape Analysis Reveals That Involvement of Pituitary Level Transcription Factors Modulate Incubation Behaviors of Magang Geese"

_genes, 2023, doi:10.3390/genes14040815_

Round 1

Reviewer 1 Report

This review pertains to the manuscript “Transcriptomic and chromatin landscape analysis reveals that involvement of the transcription factors modulate incubation behavior of Magang geese in pituitary level”. A few comments are mentioned below.

1. The Abstract is clearly and accurately describe the content of the article, but it was too long. I suggests to be summarized 

2. For products please be consistent when writing name, city, state, country through all the manuscript

3. The length of article is adequate

4. The number of tables is adequate

5. The number of figures is adequate

6. The quality is good

7. The originality is average

8. Overall:  Excellent

Author Response

Dear Reviewer 1,

Thank you very much for recognizing the insights of our studies and providing further suggestions to improve our manuscript. Following your suggestions, we rephrased the abstract section in the new edition of the manuscript and added the information for products in the Materials and Methods section.

We have listed the point-by-point replies below and incorporated our replies in the new edition of the manuscript. In the resubmitted manuscript, all the revisions have been marked with tracks.

Thank you very much for your valuable comments on our manuscript.

Sincerely yours,

Xu Shen

Associate professor of College of Animal Science & Technology,

Zhongkai University of Agriculture and Engineering,

Guangzhou 510225, China.

Email: shenxu@zhku.edu.cn

  1. The Abstract is clearly and accurately describe the content of the article, but it was too long. I suggests to be summarized.

Reply and Revision-:Thank you very much for your suggestions. We have summarized and rephrased the abstract section in the new edition of the manuscript which now reads

“The incubation behavior of geese seriously affects their egg production performance. Studies on incubation behavior have identified functional genes, but the regulatory architecture relationship between functional genes and chromatin accessibility remains poorly understood. Here, we pre-sent an integrated analysis of open chromatin profiles and transcriptome to identify the cis-regulatory element and their potential transcription factors involved in regulating incubation behavior in goose pituitary.

Assay for Transposase-accessible chromatin sequencing (ATAC-seq) revealed that open chromatin regions increased in the pituitary during the transition from incubation behavior to laying. We identified 920 significantly differential accessible regions (DARs) in the pituitary. Compared to the laying stage, most DARs had higher chromatin accessibility in the brooding stage. Motif analysis of open DARs showed that the most significant transcription factor (TF) occupied sites predominantly enriched in motifs binding to the RFX family (RFX5, RFX2, and RFX1). While the majority of TF motifs enriched under sites of nuclear receptors (NR) family (ARE, GRE, and PGR) in closed DARs at the incubation behavior stage. Footprint analysis indicated that the transcription factor RFX family exhibited higher binding on chromatin at the brooding stage. To further elucidate the effect of changes in chromatin accessibility on gene expression levels, a comparison of the transcriptome revealed 279 differentially expressed genes (DEGs). The transcriptome changes were associated with processes of steroid biosynthesis. By integrating ATAC-seq and RNA-seq, few DARs directly affect incubation behavior by regulating the transcription levels of genes. Five DAR-related DEGs were found to be closely related to maintaining the incubation behavior in geese. Footprinting analysis revealed a set of transcription factors (RFX1, RFX2, RFX3, RFX5, BHLHA15, SIX1, and DUX) displayed the highest activity at the brooding stage. SREBF2 was predicted to be the unique differentially expressed transcription factor whose mRNA level was down-regulated and enriched in hyper-accessible regions of PRL in the broody stage.

In this present study, we comprehensively profiled the transcriptome and chromatin accessibility in the pituitary related to incubation behavior. Our finding provided insight into the identification and analysis of regulatory elements in geese incubation behavior. The epigenetic alterations profiled here can help decipher the epigenetic mechanisms that contribute to the regulation of incubation behavior in birds.”

  1. For products please be consistent when writing name, city, state, country through all the manuscript

Reply and Revision-:Thank you for pointing this out. We have supplemented the information requested for the products in the Material and Method section in the new edition of the manuscript.

Reviewer 2 Report

The article submitted for review, entitled "Transcriptomic and chromatin landscape analysis reveals that involvement of the transcription factors modulate incubation behaviour of Magang geese in pituitary level", concerns the identification of genetic factors determining the behaviour of egg incubation ability in geese.

The experiments were conducted in accordance with the art of conducting experimental experiments. The techniques used are not objectionable. No overrun of auto-citations was observed.

A great deal of work was done and the results are very interesting.

However, a few remarks come to mind when reading the manuscript:

1. Figures 2, 3, 4 are completely illegible. The letters are either microscopic in size or blurred. In fact, in this form they add nothing. Please improve the sharpness and appearance of the figures, especially in view of figure 5 which is unnaturally huge in relation to its content.

2. The reviewer has doubts about the sense of the research undertaken. While from the cognitive point of view this research is interesting, from the application point of view it is of marginal importance. According to the authors, how should the identification of genetic factors determining the behaviour of the instinct to lay eggs affect bird breeding? In industrial practice, eggs are incubated in incubators where conditions are strictly controlled. In the reviewer's opinion, the presented research has no application value. Therefore, both the aim, description and summary are incomplete and do not reflect the nature of the research conducted. 

For this reason, the article as presented is not suitable for publication and should be remodelled to take into account the nature of the research and its possible use.

Author Response

Dear Reviewer 2,

Thank you very much for recognizing the insights of our studies and providing further suggestions to improve our manuscript. Following your suggestions, we reorganized the figures to enhance the quality of the figures and explained the sense, and summarized the application value of our research.

We have listed the point-by-point replies below and incorporated our replies in the new edition of the manuscript. In the resubmitted manuscript, all the revisions have been marked with tracks.

Thank you very much for your valuable comments on our manuscript.

Sincerely yours,

Xu Shen

Associate professor of College of Animal Science & Technology,

Zhongkai University of Agriculture and Engineering,

Guangzhou 510225, China.

Email: shenxu@zhku.edu.cn

Reviewer 2:

1.Figures 2, 3, 4 are completely illegible. The letters are either microscopic in size or blurred. In fact, in this form they add nothing. Please improve the sharpness and appearance of the figures, especially in view of figure 5 which is unnaturally huge in relation to its content.

Reply and Revision-:Thank you for the comment. For Figures 2,3, and 4, We have replaced the figures with low-quality and improved the sharpness and appearance of the figures. For Figure 5, we adjusted the size and improved the resolution.

  1. The reviewer has doubts about the sense of the research undertaken. While from the cognitive point of view this research is interesting, from the application point of view it is of marginal importance. According to the authors, how should the identification of genetic factors determining the behaviour of the instinct to lay eggs affect bird breeding? In industrial practice, eggs are incubated in incubators where conditions are strictly controlled. In the reviewer's opinion, the presented research has no application value.Therefore, both the aim, description and summary are incomplete and do not reflect the nature of the research conducted.

Reply and Revision-:Thank you for raising this issue. We are gracefully to explain the sense and the application value of our research.

  1. the sense of the presented research

A low laying performance in the goose caused by strong incubation behavior is one of the bottlenecks preventing the development of the goose industry. Magang geese have a strong tendency to incubation behavior, Incubation behavior leads to ovarian regression and the termination of ovulation[1] (shown in Figure R1 A, B). 90% of Magang geese exhibited incubation behavior after laying one clutch of approximately eight eggs in approximately 30 days, and the incubation duties of the eggs last for 40 days before they hatch[2]. As a result, incubation behavior limits the annual egg production of Magang geese capacity to approximately 30-40 eggs, with less than 30 goslings hatched[3], which causes significant economic losses in the geese industry.

Figure R1 Ovarian morphology of Laying geese and brooding geese. (A) Morphology characteristics of the ovary in egg-laying geese and (B) the regression ovary in broody geese

  1. The application value of the presented research

Incubation behavior is a special reproductive behavior formed in the long-term evolutionary process of birds. Incubation behavior in poultry is characterized by persistent nesting, accompanied by atrophy of the ovaries and oviduct and the eventual termination of egg production.

In the modern poultry industry, with the development of artificial incubation technology, incubation behavior is no longer needed in poultry production. Incubation behavior has become an important factor restricting the laying performance and feeding efficiency of poultry. Therefore, the reduction and elimination of incubation behavior have become one of important breeding objectives. Although traditional breeding efforts have made considerable genetic progress in controlling incubation behavior in poultry due to the low heritability of incubation behavior, the potential for such improvement is diminishing, but it is difficult to completely eradicate.

Incubation behavior still exists in some indigenous chicken breeds and goose breeds. The information derived from our study could be valuable for understanding the genetics of incubation behavior and could facilitate further investigation into breeding against broodiness.

  1. Liu L, Xiao Q, Gilbert ER, Cui Z, Zhao X, Wang Y, Yin H, Li D, Zhang H, Zhu Q: Whole-transcriptome analysis of atrophic ovaries in broody chickens reveals regulatory pathways associated with proliferation and apoptosis. Sci Rep 2018, 8(1):7231.
  2. Huang YM, Shi ZD, Liu Z, Liu Y, Li XW: Endocrine regulations of reproductive seasonality, follicular development and incubation in Magang geese. Animal Reproduction Science 2008, 104(2):344-358.
  3. Qin Q, Sun A, Guo R, Lei M, Ying S, Shi Z: The characteristics of oviposition and hormonal and gene regulation of ovarian follicle development in Magang geese. Reproductive Biology and Endocrinology 2013, 11(1):65.

Round 2

Reviewer 2 Report

Thank you for your explanation no further comment